# Validation and Implementation of OptiView and EnVision FLEX Detection Systems for Immunocytochemical Staining Protocols of the Ten Most Commonly Used Diagnostic Markers in Routine Cytopathological Practice

**DOI:** 10.3390/diagnostics14060657

**Published:** 2024-03-21

**Authors:** Anja Dremelj, Simona Miceska, Anamarija Kuhar, Natasa Nolde, Veronika Kloboves-Prevodnik

**Affiliations:** 1Institute of Oncology Ljubljana, Zaloska cesta 2, 1000 Ljubljana, Sloveniasmiceska@onko-i.si (S.M.);; 2Faculty of Pharmacy, University of Ljubljana, Askerceva 7, 1000 Ljubljana, Slovenia; 3Faculty of Medicine, University of Ljubljana, Vrazov trg 2, 1000 Ljubljana, Slovenia; 4Faculty of Medicine, University of Maribor, Taborska ulica 8, 2000 Maribor, Slovenia

**Keywords:** BenchMark ULTRA immunostainer, cytopathology, Dako Omnis immunostainer, EnVision FLEX, immunocytochemistry, iView, OptiView, detection systems, staining protocols

## Abstract

The withdrawal of the iView detection system (iV) forced many cytopathology laboratories, including ours, to substitute immunocytochemical (ICC) staining protocols for routine practice with other detection systems. Our objective was to optimize, validate, and implement ICC protocols using OptiView (OV) and EnVision FLEX (EnV) detection systems, comparing the results with those obtained using iV. Residual cytologic samples with known diagnoses were used, testing antibodies for the ten most common markers in routine cytopathology diagnostics (calretinin, Ber-EP4, MOC-31, CKAE1/AE3, CK5/6, CD68, LCA, desmin, HBME-1, and WT1). Different staining parameters were tested using OV on BenchMark ULTRA and EnV on Dako Omnis immunostainer, respectively. Optimal staining protocols were then selected and validated on 10 positive and 10 negative cases. The staining results were compared with iV protocols through evaluation of UK NEQAS and internal scores. The optimal staining protocols with OV and EnV demonstrated similar or superior results compared to the existing iV protocols, with slightly stronger intensity regarding positive cells. We have successfully established and validated optimal ICC staining protocols for commonly used markers in routine cytopathology practice. These protocols may benefit other laboratories using similar staining platforms. However, the challenge regarding standardizing ICC protocols across different cytopathology laboratories remains unresolved.

## 1. Introduction

Immunocytochemistry (ICC) is still an indispensable additional technique to identify the cell lineage and origin of tumors from cytological samples but is unfortunately less standardized than immunohistochemistry (IHC) from tissue samples [1,2,3]. Therefore, cytopathology laboratories face the issue of adjusting and validating their ICC staining protocols for diagnostic markers according to the available IHC reagents and immunostainers, although no standardized protocols have been established for ICC [4]. Many laboratories solve this problem by using histology protocols on cell blocks [5,6,7,8]. However, cell blocks are not a solution for all cytology samples, especially those containing only a few cells. For such samples, ICC staining on cytospins is the more appropriate technique. Routine cytopathology practice at the Institute of Oncology Ljubljana (IOL), Ljubljana, Slovenia, has shown that ICC staining for diagnostic markers on methanol-fixed cytospins using the iView (iV) detection system (Ventana, Roche Diagnostics, Rotkreuz, Switzerland) provides very good results [9,10,11]. Consequently, it has been widely used for many years not only in our laboratory but also in other cytopathology laboratories in Slovenia [12,13,14,15,16] and abroad [17,18,19]. Since the iV detection system has been recently withdrawn from the market, with reagents still available until 2023, many cytopathology laboratories, including ours, were forced to find a replacement in terms of another detection system and adjust our ICC staining protocols. Therefore, the objective of our study was to optimize, validate, and implement ICC staining protocols for the ten most commonly used markers for routine cytopathological diagnostics with the OptiView (OV) and EnVision FLEX (EnV) detection systems (Roche Diagnostics, Rotkreuz, Switzerland, and Dako Omnis, Agilent Technologies, Santa Clara, CA, USA, respectively) by comparing the staining results with those obtained with the existing iV detection system.

## 2. Materials and Methods

### 2.1. Patients

The study included patients who underwent effusion puncture, abdominal washing, or fine needle aspiration biopsy (FNAB) as part of the diagnostic procedure between March and June 2022. The cytology samples were sent to the Department of Cytopathology at the IOL for routine cytopathology evaluation, and only samples’ leftovers were used for this study, considering both non-neoplastic diseases (atypical mesothelial proliferation, inflammation, and adenocarcinomas without tumor cells in the effusions) and malignant neoplasms (lung-, breast-, and ovarian adenocarcinomas and non-Hodgkin’s lymphomas). The inclusion criteria were as follows: an authorized cytopathology report, sufficient residual material of the sample, and approval from an experienced cytopathologist (VKP) for which the analyzed diagnostic markers and/or as a negative control the sample should be used (according to the previously established diagnosis and the expected positive and/or negative staining result).

### 2.2. Study Design

The antibodies against calretinin, Ber-EP4, MOC-31, CK AE1/AE3, CK 5/6, CD68, LCA, desmin, HBME-1, and WT1 (Table 1), which are most frequently used diagnostic markers in our routine cytopathological practice, were selected for the validation study, which consisted of two parts (Figure 1). In the first part, we tested different parameters (antibody dilution and incubation time, antigen (target) retrieval, amplification, and counterstaining) for OV and EnV detection systems to select a unique optimal staining protocol by comparing the staining results with routinely used iV staining protocols at the IOL on five different positive and negative samples for each antibody separately. The protocol that provided the best results was selected for further evaluation in the second part of the study. The selected protocols were validated on 10 positive and 10 negative samples with known diagnoses according to previously published validation procedures [12,20]. The results obtained with the OV and Env detection systems were compared with the results of the iV detection system used in routine diagnostics. The internal IOL and UK NEQAS (NEQAS) scoring systems were used to evaluate the ICC staining results and for further statistical analysis using intraclass correlation and kappa statistics.

### 2.3. Cytological Sample Preparation

Giemsa and Papanicolaou stained smears for each sample were prepared as a part of the routine diagnostics at IOL. Furthermore, methanol-fixed cytospins with optimal cellularity and quality were prepared as previously described by our group [11]. The number of cytospins prepared from one sample varied among different samples due to the differences in the number of cells in them. The cytospins were stored in methanol solution at 4 °C up to ICC staining.

### 2.4. Immunocytochemical Staining

ICC staining with iV and OV detection systems was performed on the BenchMark ULTRA immunostainer (Ventana, Roche Diagnostics, Rotkreuz, Switzerland), ICC staining with EnV on the Dako Omnis immunostainer (Agilent Technologies, Santa Clara, CA, USA). To optimize the OV and EnV protocols, different antibody dilutions and different incubation times (8, 16, and 32 min) were tested for each antibody. If the results with these optimization parameters were not satisfactory, a target retrieval step was added. For OV Cell Conditioning, Reagents 1 and 2 (CC1 and CC2) (Ventana, Roche Diagnostics, Rotkreuz, Switzerland) were used and, for EnV Target Retrieval Solution, low pH, Target Retrieval Solution, and high pH (Agilent Technologies, Santa Clara, CA, USA). For the test protocols in which the positive staining was still weak after target retrieval, amplification with the OptiView Amplification Kit (Ventana, Roche Diagnostics, Rotkreuz, Switzerland) or the Mouse Linker Reagent (Agilent Technologies, Santa Clara, CA, USA) was added for OV and EnV, respectively. A detailed description of the parameters tested for each detection system and antibody is provided in Table 2. The detection systems differed in the mechanism regarding how a positive reaction was obtained (Figure 2), but all three used diaminobenzidine (DAB) as a chromogen, resulting in a brown precipitate as positive reaction.

### 2.5. Microscopic Evaluation

The ICC staining results were evaluated semi-quantitatively by four independent investigators (A.D., A.K., N.N., and V.K.P.). For the first part of the study, all investigators reached a consensus in selecting the optimal protocol for each marker using OV and EnV detection systems under a multi-head microscope, and, for the second part, each investigator provided an individual assessment result. Scoring was completed by IOL and NEQAS scores (the IOL score differs from the NEQAS score by the inclusion of two additional parameters: percentage of positive cells and preservation of cell morphology).

### 2.6. Scoring Criteria

According to the IOL scoring criteria, the staining was assessed as 0 (no positive staining), 1 (<15% stained cells), 2 (15–50% stained cells), and 3 (>50% stained cells). Intensity was scored with 0 (no staining), 1 (weak), 2 (moderate), and 3 (strong). Background was scored with 0 (no background), 1 (weak), 2 (moderate), and 3 (strong). The type of background was additionally categorized as nuclei, histocyte cytoplasm, another cell cytoplasm, or extracellular matrix. A score of 0 was used for appropriate counterstaining and preserved cell morphology, while 1 indicated inappropriate counterstaining and altered cell morphology [16]. According to the NEQAS scheme, slides were first scored from 0 to 5 by each of the investigators individually, and the score was added up to an overall score, which provided information on the quality of the stained slide. The score was as follows: lower than 8/20 (unacceptable quality), 9–11/20 (consensus was needed to choose between unacceptable or marginally acceptable quality), 12/20 (marginally acceptable), 13–15/20 (acceptable), and higher than 16/20 (good to excellent quality) [18].

### 2.7. Statistical Analysis

Inter-rater agreement was calculated using intraclass correlation and kappa coefficients for NEQAS score, staining, intensity, background, counterstaining, and morphology. Intraclass coefficient was categorized as poor (0–0.50), moderate (0.50–0.75), good (0.75–0.90), and excellent (0.90–1.00) agreement [21]. Kappa coefficient was assessed as no agreement (≤0), none to slight (<0.20), sufficient (0.21–0.40), moderate (0.41–0.60), substantial (0.61–0.80), and (almost) perfect agreement (0.81–1.00) [22]. One-factor ANOVA was used to compare NEQAS scores and staining between all three detection systems, Friedman’s test to compare intensity and background, and Cochran test to compare counterstaining and morphology. For the significantly different results between abovementioned variables, pairwise *t*-test, Wilcoxon rank test, and McNemar test were further used. *p* < 0.05 was considered as significant. Statistical analysis was performed in RStudio v2022.07.01.

## 3. Results

### 3.1. Cytological Samples

Samples of 52 patients, of whom 31 were diagnosed with non-neoplastic disease (atypical mesothelial proliferation, inflammation, or adenocarcinomas without tumor cells in the effusions) and 21 with malignant neoplasms (lung-, breast-, and ovarian adenocarcinomas and non-Hodgkin’s lymphomas), were included in the study. Of all fifty-two patients’ samples, forty-three were effusions (thirteen abdominal effusions, twenty-nine pleural effusions, and one effusion in the pouch of Douglas), two were washings from the pouch of Douglas and one from the abdomen, and six were FNABs. Due to the different biological characteristics of each sample, varying numbers of cytospins were prepared from the different samples. Detailed descriptions of the samples’ types and origins are provided in Table 3.

### 3.2. Selection of Optimal Staining Protocols with OptiView and EnVision FLEX Detection Systems

The staining protocols used for the optimization of the OV and EnV detection systems varied among the tested antibodies against calretinin, Ber-EP4, MOC-31, CK AE1/AE3, CK 5/6, CD68, LCA, desmin, HBME-1, and WT1 due to the different characteristics of the detection systems and IHC platforms used (Figure 3 and Figure 4).

Staining with the OV detection system for calretinin provided non-satisfactory results if additional steps of target retrieval were not included in the staining protocol regardless of the antibody incubation time (8, 16, or 32 min). However, including a target retrieval step with the CC1 solution minimized the nonspecific staining of the samples, whereas an amplification step improved the intensity of the positive stained cells. Similarly, the most optimal staining results of Ber-EP4 and MOC-31 were achieved under the same protocol conditions as for calretinin. The optimal staining intensity and the absence of background staining of CK AE1/AE3, CK 5/6, and HBME-1 were achieved using the protocol with CC1 target retrieval accompanied by 16 min, 32 min, and 32 min antibody incubation time, respectively. For CD68, LCA, and desmin detection, 32 min antibody incubation without additional steps was chosen as the optimal protocol condition. For WT1 detection, the amplification step optimized the staining intensity and specificity of the previously routinely used protocol with target retrieval with the CC1 solution. Target retrieval using CC2 did not yield satisfactory results for any of the test protocols and antibodies. The selection of optimal protocol conditions is provided in Table 2 (bolded parameters).

The staining results with the EnV detection system for calretinin were satisfactory even without the need to include target retrieval, while, for Ber-EP4 and MOC-31, target retrieval in low-pH solution showed the best staining intensity and almost no background staining. Strong staining intensity of CK AE1/AE3 and LCA was achieved without target retrieval and 5 min and 30 min antibody incubation, respectively. However, some background staining was still present. The most optimal staining results for CD68 and HBME-1 required target retrieval in low-pH solution, while for desmin that step was not necessary since the best result was achieved without target retrieval. Satisfactory results regarding CK 5/6 and WT1 staining were obtained when adding a target retrieval step in a high-pH solution at 97 °C. Moreover, the additional inclusion of Mouse Linker to amplify the signal was essential to improve the staining intensity of WT1. The selection of optimal protocol conditions is provided in Table 2 (bolded parameters).

### 3.3. Comparison of OptiView and EnVision FLEX Optimal Staining Protocols with the Existing iView Protocol

In Table 4, we present the results of the OV and EnV staining protocols, which were selected in the first part of the study, and the results of the routinely used iV staining protocols. The median values for the staining results (percentage of positive cells), intensity of ICC reaction, background, counterstain, and morphology assessment obtained by four independent investigators are presented for 10 samples with expected positive ICC reaction. The agreement between all four investigators was good. They obtained excellent agreement for the staining results (k = 0.961), moderate agreement for the intensity, background, and counterstaining criteria (k = 0.441, k = 0.506, and k = 0.472, respectively), and fair agreement for the morphology assessment (k = 0.300).

The results of the comparison of the optimal protocols of the OV, EnV, and iV detection systems for all 10 diagnostic markers, according to the IOL and NEQAS evaluation criteria, are shown in Table 5; representative images of the stainings are provided in Figure 5. The chosen optimal staining protocols with OV and EnV for calretinin showed better quality assessed by NEQAS score (*p* < 0.001 and *p* = 0.002, respectively) compared to the existing iV protocol. Furthermore, the intensity of positive cells was stronger with OV and EnV (*p* = 0.003, *p* = 0.031). The results of Ber-EP4 and MOC-31 detection with OV in comparison with iV resulted in better NEQAS scores (*p* = 0.001, *p* = 0.017) and stronger staining intensity (*p* = 0.012, *p* = 0.007). The quality of the stained slides with EnV was comparable to iV since there were no significant differences in NEQAS score, staining, intensity, background, counterstain, and morphology. The OV and EnV staining protocols for CK AE1/AE3 revealed higher NEQAS scores (*p* = 0.004, *p* = 0.005) and intensity (*p* = 0.004, *p* = 0.012) compared to iV.

However, counterstaining with EnV was less intense when compared to OV. The OV staining protocol for CK 5/6 showed the highest NEQAS score (*p* = 0.002) compared to the other two protocols. However, the intensity with OV and EnV was better than with iV (*p* = 0.006 and *p* = 0.012, respectively). For CD68, no significant difference was observed between the protocols with all three detection systems. Furthermore, we observed that the iV staining protocol for LCA did not stain all the leukocytes in the corresponding cytospins, whereas OV and EnV did. The intensity of the stained cells with OV and EnV was stronger than with iV (*p* = 0.005, *p* = 0.007). For desmin, the staining and intensity of the stained slides were significantly better using OV (*p* = 0.029, *p* = 0.013) compared to iV; however, more nonspecific reactions were observed for the detection of this marker with EnV. Protocols for detection of HBME-1 exhibited comparable results, except for intensity, which was stronger with OV (*p* = 0.006), and NEQAS score, which was higher with EnV (*p* = 0.019). The staining for WT1 showed the highest NEQAS score with the OV protocol (*p* = 0.007, *p* = 0.002) and significantly stronger staining intensity compared to the iV and EnV protocols (*p* = 0.003 and *p* = 0.004, respectively), while the background was still present when using the EnV protocol.

In all three detection systems, a nonspecific background was observed in both positive and negative samples. Background staining for the OV detection system was mainly observed in the cytoplasm of macrophages and was reduced by the introduction of target retrieval with CC1. However, the intensity of the staining reaction was weak; therefore, amplification of the signal was required. In the EnV detection system, background staining was also observed in macrophages and additionally in the cytoplasm of other cells, including erythrocytes and extracellular matrices, but was successfully reduced by establishing high pH protocols. For example, for WT1 detection, the OV protocol with amplification improved the same OV protocol without the amplification as the NEQAS score and intensity results were significantly better, and the background staining was reduced (*p* = 0.007). However, the EnV protocol for WT1 resulted in more nonspecific staining, specifically on the erythrocytes. The counterstaining and morphology assessment of calretinin, Ber-EP4, MOC-31, CK AE1/AE3, CK 5/6, CD68, LCA, desmin, HBME-1, and WT1 did not show any differences among the detection systems. Furthermore, no false-positive staining was observed on the samples used as negative controls.

## 4. Discussion

There are no standardized protocols for ICC staining on cytospins, yet, in our routine cytopathology practice at the IOL over many years, we have obtained satisfactory results with the iV detection system [7,9,11], which has recently been withdrawn from production. Here, we tested two other IHC detection systems, OV and EnV, and the 10 most commonly used diagnostic markers in routine cytopathology diagnostics and observed similar or even better staining results compared to iV.

The withdrawal of the iV detection system from the market lacks a specific explanation. However, based on our comprehensive literature review, it appears that IHC, which is more prevalent than ICC and has established standardization, has favored the use of the OV detection system over iV. This preference has been mainly attributed to the introduction of unique biotin-free synthetic HQ haptens by OV instead of a biotin–streptavidin system as in iV, where the endogenous biotin itself might result in nonspecific staining if it is not blocked properly (Figure 2). Moreover, more sensitive and specific staining achieved with OV rather than iV has been reported [23]. On the other hand, the EnV detection system utilizes a dextran-polymer-based secondary antibody system, again avoiding biotin-related nonspecificity (Figure 2) [23]. However, it is worth noting that the primary drawback of EnV could be its higher cost when compared to alternatives like iV and OV [23].

The main aim of our study was to create optimal ICC staining protocols with the OV detection system on BenchMark ULTRA immunostainer and the EnV system on Dako Omnis immunostainer for the 10 most frequently used diagnostic markers in our cytopathology laboratory, validate them, and implement them in clinical practice. To the best of our knowledge, no prior cytological study has undertaken a comparative investigation involving these three detection systems. Consequently, we were unable to compare the outcomes of our study with other data. This observation further underscores the prevailing practice in cytopathological laboratories, where histology protocols are used on cell blocks, or protocols for ThinPrep or SurePath samples for liquid-based Papanicolaou tests are often created and adjusted exclusively for in-house use, and data sharing or publication remain unknown [24]. It is important to highlight that, even in histology, the choice of different detection systems can significantly influence the final staining outcomes. Indeed, Skaland et al. have published variations in sensitivity, specificity, and costs [25]. Noteworthily, their study compared different detection systems in histology, but, except for the EnV detection system, they included four other detection systems that were not a part of our study.

However, our practice through the study showed that combining different parameters within protocols using different detection systems could lead to satisfactory ICC staining protocols for routine cytopathological practice, depending on the available immunostaining equipment. Furthermore, our long experience with ICC at the IOL [26] allowed us to properly validate the selected optimal protocols with each antibody used in the study in accordance with the validation protocols published previously. Good optimization and validation of protocols are essential to maintain the quality, reliability, and reproducibility of staining results in clinical practice and to facilitate the process of standardization in the near future [20,27,28].

Because we based our study on comparing our already existing protocols on methanol-fixed cytospins using the iV detection system with protocols using the OV and EnV detection systems, we noticed that the majority of the OV-based protocols required target retrieval with CC1, which was not the case with the iV detection system. CC1 is a commercial retrieval reagent by Roche Diagnostics (containing TRIS/EDTA buffer, pH 7.8), indicating that, due to the complexity of the OV detection system, which is characterized by antibodies and HQ linkers binding, proper exposure of antigenic epitopes and improved binding of antibodies are essential (Figure 2). Another advantage of OV is the inclusion of an amplification step with the OV Amplification kit, which is a tyramide-based reagent and facilitated positive reactions in four of the tested antibodies [29]. There are also reports in IHC, where optimal results were more frequently achieved with the addition of the Amplification kit [30,31,32]. Moreover, our group has recently published a study on cytospins and cell blocks from cytopathology effusions using the OV detection system, where the staining results were compared with those of the corresponding primary tumors, indicating a high level of reliability [11]. These findings confirm that, even though the IHC staining is adjusted for histology, good laboratory practices can yield cytopathology staining results as reliable as those from histology biopsies, and also indicate that in cytology cell blocks could facilitate the standardization of ICC staining protocols rather than cytospins. However, for many antibodies frequently used in routine diagnostics, satisfactory staining can also be obtained with cytospins, which is important for samples that contain only a few tumor cells and therefore cell blocks cannot be prepared [6,33]. This allows immunophenotypic characterization of tumor cells. For EnV, target retrieval was required in the majority of the optimal protocols, similar to OV, referring to the complexity of the detection system and better exposure of the target antigens, and there was almost no need for amplification with the Mouse Linker. According to the NEQAS scores, using optimal ICC staining protocols performed by the OV and EnV detection systems, stronger intensity was achieved, even though EnV was characterized by less intense counterstaining. We speculate that these improved results could be attributed to the detection mechanism not relying on streptavidin–biotin interactions [34]. As we could not entirely eliminate background staining in all three detection systems, we suspect that this phenomenon might be related to the nature of cytological samples and the different preparation stages, such as sample collection timing, fixation, and staining intervals. To note, the drawback of the OV and EnV protocols is the extended time of the ICC staining due to the inclusion of target retrieval and amplification steps, which influence the workflow efficiency in cytopathology laboratories since their advantage is fast examination, diagnosis, and treatment decisions [29]. These additional steps also affect the price of the performed testing.

However, our study had a few limitations. As it was committed to good ethical practice to avoid taking additional samples from patients [35], we only used leftover samples from patients whose samples had originally been taken for routine diagnostic purposes at the IOL, and the majority of the samples were effusions rather than FNAB samples from lymph nodes. The main reason for this was the greater availability of leftover cellular material from the effusions, which allowed us to produce a greater amount of additional cytospins to perform the tests, compared to FNAB samples, where in most cases the entire sample was used to complete the diagnostic testing, as is the case with other laboratories [4,36], or flow-cytometric analysis is preferred to establish a lymphoma diagnosis [37]. These restrictive conditions might possibly raise concerns about the reliability of our staining protocols for the same markers in samples of different origin. However, our long-term experience [26] and analysis of follow-up data at the IOL have confirmed that the application of optimized protocols for samples of different origin does not lead to variations in the staining reaction (unpublished data). This action is controlled by a positive and negative control cytospin for each marker in each staining round [7,11,16,38]. Another limitation of our study is the limited number of diagnostic markers included in the analysis, but, again, our main objective was to compare the OV and EnV detection systems with iView to investigate whether we can replace our staining protocols with other detection systems and staining platforms designed primarily for histology, and that is a worthy cost-benefit ratio [29,39]. However, considering the good comparability of all detection systems, our future goal is to extend the optimization and validation for the other more rarely used diagnostic markers in cytopathology and publish the results to impact the process of protocol standardization in all cytopathology laboratories [39].

## 5. Conclusions

To conclude, we created and validated optimal ICC staining protocols for the ten most commonly used diagnostic markers in routine cytopathology practice. These protocols were developed for use with OV and EnV on both BenchMark ULTRA and Dako Omnis immunostainers, respectively. In comparison to the existing routine staining protocols with the iV detection system, which was recently withdrawn from production, we achieved similar or even better staining quality, enabling their integration into clinical practice. These protocols might be useful for other cytopathology laboratories with the same immunostainers. However, the challenge of standardizing ICC protocols across cytopathology laboratories remains unresolved.

## Figures and Tables

**Figure 1 diagnostics-14-00657-f001:**
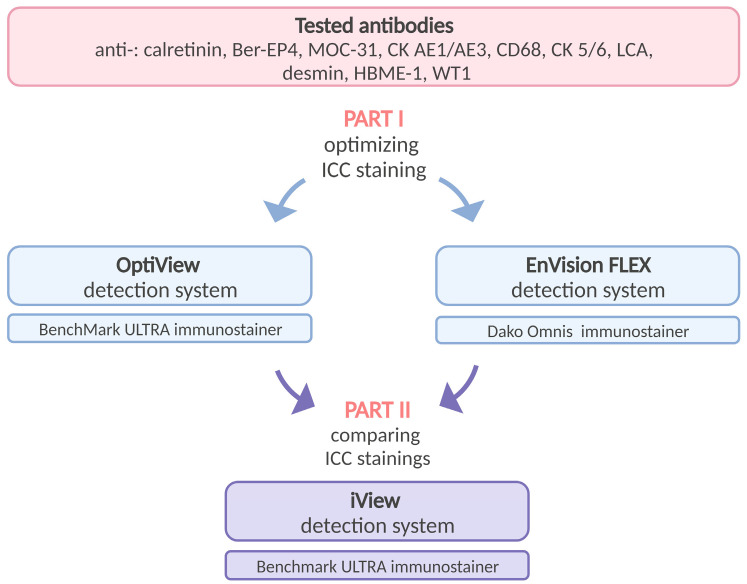
Schematic description of the study design. Created with Biorender.com.

**Figure 2 diagnostics-14-00657-f002:**
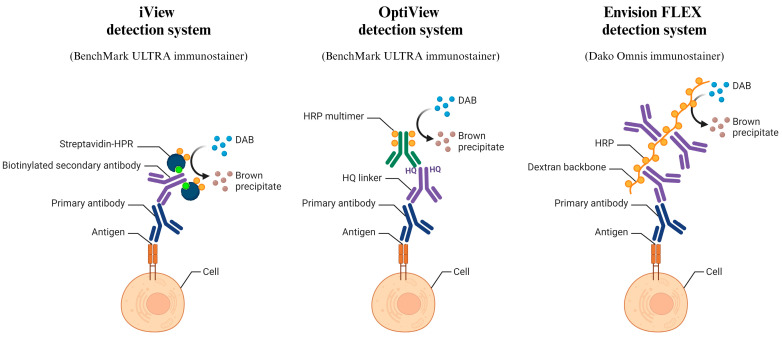
The iView (iV), OptiView (OV), and EnVision FLEX (EnV) detection systems use different mechanisms. iV is based on a biotin–streptavidin reaction, OV uses HQ linkers, and EnV uses a dextran backbone, all leading to the formation of the final diaminobenzidine (DAB) precipitate as a positive staining reaction. iV and OV detection systems are specifically designed for the BenchMark ULTRA, while EnV is optimized for the Dako Omnis immunostainer. Created with Biorender.com.

**Figure 3 diagnostics-14-00657-f003:**
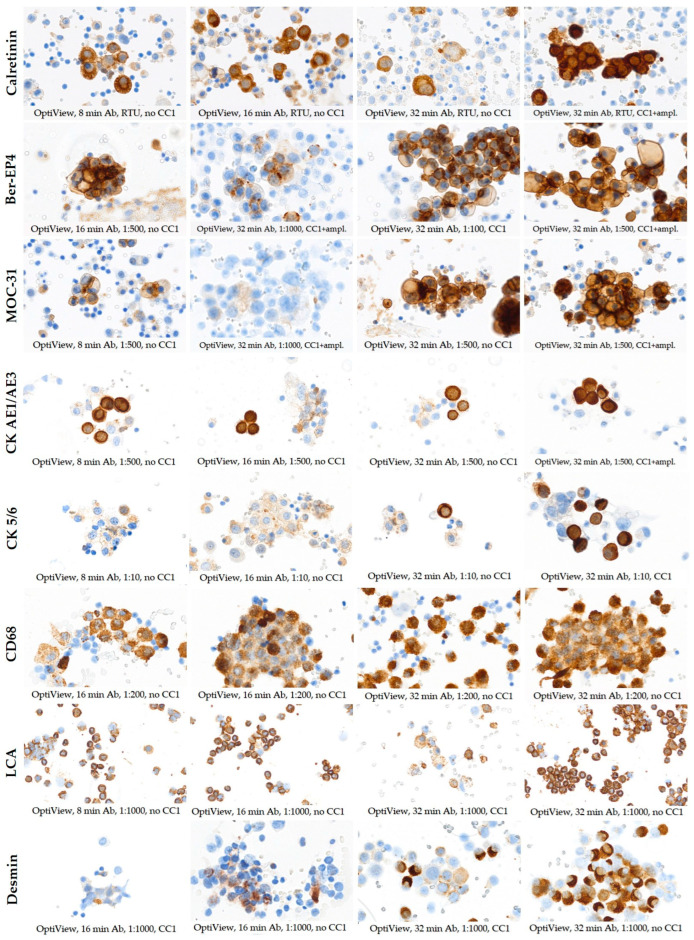
Immunocytochemical staining results for calretinin, Ber-EP4, CK AE1/AE3, CK 5/6, CD68, LCA, desmin, and WT1 with OptiView detection system using different test protocols (×400 magnification). The conditions of each test protocol are labeled directly under the corresponding image within the figure.

**Figure 4 diagnostics-14-00657-f004:**
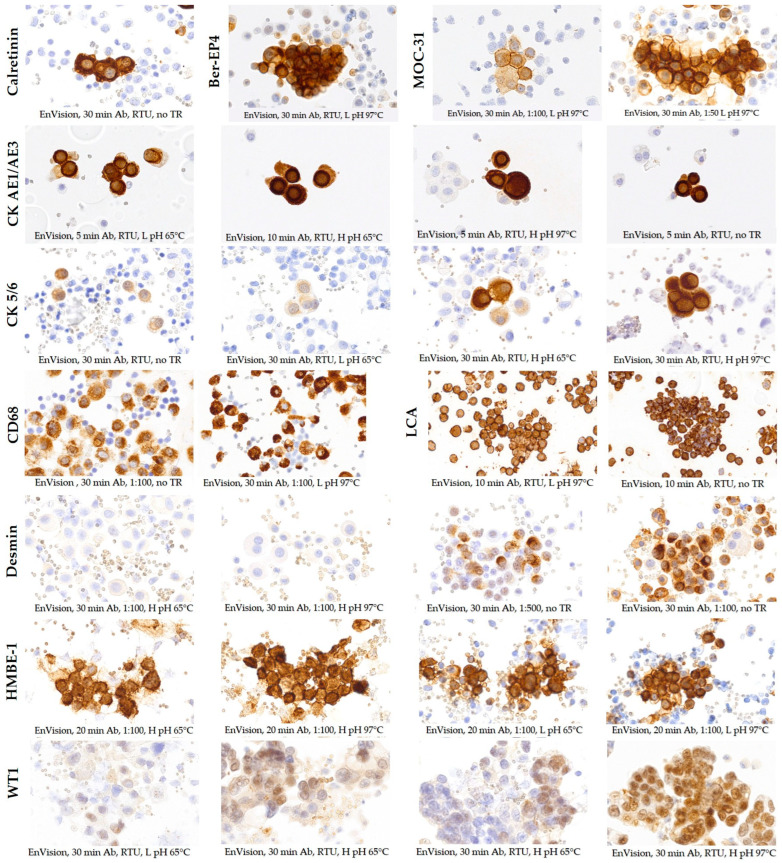
Immunocytochemical staining results for calretinin, Ber-EP4, CK AE1/AE3, CK 5/6, CD68, LCA, desmin, and WT1 with EnVision FLEX (EnVision) detection system using different test protocols (×400 magnification). The conditions of each test protocol are labeled directly under the corresponding image within the figure.

**Figure 5 diagnostics-14-00657-f005:**
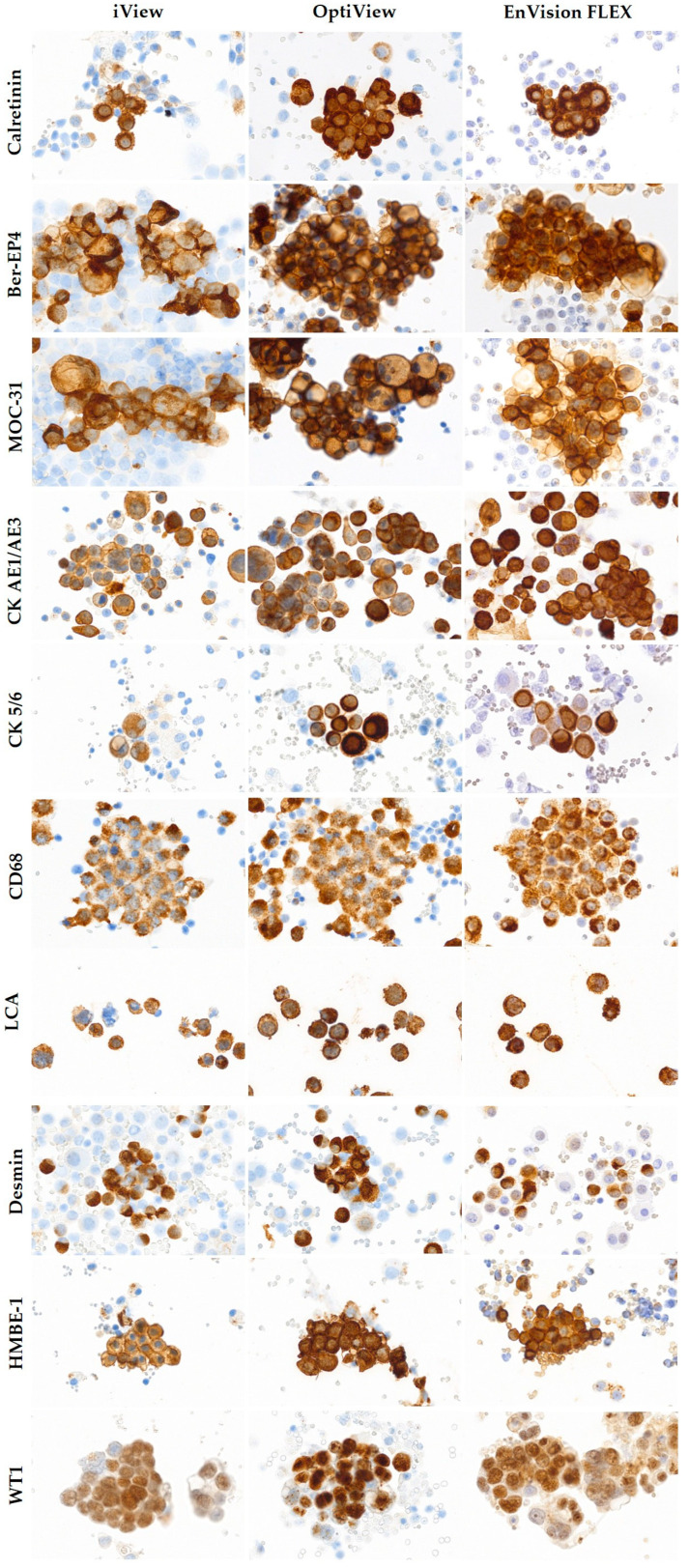
Comparison of the selected OptiView and EnVision FLEX immunocytochemical staining protocols for calretinin, Ber-EP4, CK AE1/AE3, CK 5/6, CD68, LCA, desmin, and WT1 with the existing iView protocol (×400 magnification).

**Table 1 diagnostics-14-00657-t001:** Detailed description of the antibodies used for the tested diagnostic markers.

Diagnostic Marker	Immunostainer	Clone	Manufacturer	Localization	Target Cell for Positive Reaction
Calretinin	BenchMark ULTRA	SP4	Roche Diagnostics	cytoplasm, nucleus	mesothelial cells, mesothelioma cells
	Dako Omnis	DAK-Calret1	Agilent Technologies	cytoplasm	
Epithelial cellular adhesion molecule (EpCAM)	BenchMark ULTRA, Dako Omnis	Ber-EP4	Agilent Technologies	cytoplasm	epithelial malignant cells
EpCAM	BenchMark ULTRA, Dako Omnis	MOC-31	Dako	cytoplasm	epithelial malignant cells
Cytokeratin AE1/AE3 (CKAE1/AE3)	BenchMark ULTRA, Dako Omnis	CK AE1/AE3	Dako, Agilent Technologies	cytoplasm	epithelial cells, including normal and malignant cells
Cytokeratin CK 5/6 (CK5/6)	BenchMark ULTRA, Dako Omnis	D5/16 B4	Dako, Agilent Technologies	cytoplasm	epithelial cells, including normal and malignant cells
CD68	BenchMark ULTRA, Dako Omnis immunostainer	PG-M1	Dako	cytoplasm	macrophages
Leukocyte common antigen (LCA (CD45))	BenchMark ULTRA, Dako Omnis	2B11 + PD7/26	Dako, Agilent Technologies	membrane	immune cells
Desmin	BenchMark ULTRA, Dako Omnis	D33	Dako	cytoplasm	malignant cells with myogenic differentiation
Human bone marrow endothelium marker-1 (HBME-1)	BenchMark ULTRA, Dako Omnis	HBME-1	Dako	cytoplasm, membrane	mesothelial cells, mesothelioma cells
Wilms’ tumor protein 1 (WT1)	BenchMark ULTRA, Dako Omnis	6F-H2	Cell Marque, Dako	nucleus	epithelial cells, including normal and malignant cells

**Table 2 diagnostics-14-00657-t002:** Detailed description of the tested parameters for iView, OptiView, and Envision FLEX detection systems for each of the tested diagnostic markers. Parameters that are bolded were selected as the most optimal among other tested conditions.

Detection System	iView *	OptiView				EnVision FLEX			
Diagnostic marker	Ab dilution	Ab dilution	Ab incubation time (min)	TR	Amplification	Ab dilution	Ab incubation time (min)	Ag retrieval	Mouse Linker
Calretinin	**RTU**	**RTU**	8 16 **32**	none **CC1** CC2	**Yes**	**RTU**	**30**	**none**	**No**
Ber-EP4	**1:500**	1:25 1:50 1:100 **1:500** 1:1000	8 16 **32**	none **CC1**	**Yes**	**RTU**	**30**	**L pH, 97 °C, 10 min**	**No**
MOC-31	**1:500**	1:50 1:100 **1:500** 1:1000	8 16 **32**	none **CC1**	**Yes**	**1:50** 1:100	**30**	**L pH, 97 °C, 10 min**	**No**
CK AE1/AE3	**1:500**	**1:500**	8 **16** 32	none **CC1**	**No**	**RTU**	**5** 10	**none** L pH, 65 °C, 10 min L pH, 97 °C, 10 min H pH, 65 °C, 10 min H pH, 97 °C, 10 min	**No**
CK 5/6	**1:10**	**1:10**	8 16 **32**	none **CC1**	**No**	**RTU**	**30**	none L pH, 65 °C, 10 min L pH, 97 °C, 10 min H pH, 65 °C, 10 min **H pH, 97 °C, 10 min**	**No**
CD68	**1:200**	**1:200**	16 **32**	**none**	**No**	**1:100** 1:200	20 **30**	none **L pH, 97 °C, 10 min**	**No**
LCA	**1:1000**	**1:1000**	8 16 **32**	**none** CC1	**No**	**RTU**	**10** 20	**none** L pH, 97 °C, 10 min	**No**
Desmin	**1:1000**	**1:1000**	16 **32**	**none** CC1	**No**	**1:100** 1:500 1:1000	**30**	**none** L pH, 97 °C, 10 min H pH, 65 °C, 10 min H pH, 97 °C, 10 min	**No**
HBME-1	**1:50**	**1:50**	8 16 **32**	none **CC1** CC2	**No**	1:50 **1:100**	**20**	none L pH, 65 °C, 10 min **L pH, 97 °C, 10 min** H pH, 65 °C, 10 min H pH, 97 °C, 10 min	**No**
WT1	**1:200**	1:100 **1:500**	**32**	**CC1**	**Yes**	**RTU**	**30**	L pH, 65 °C, 10 min L pH, 97 °C, 10 min H pH, 65 °C, 10 min **H pH, 97 °C, 10 min**	**Yes**

* All protocols with the iView detection system were performed with a 32 min antibody incubation and no antigen retrieval. Abbreviations: Ab—antibody; Ag—antigen; CC1—Cell Conditioning Solution 1; CC2—Cell Conditioning Solution 2.

**Table 3 diagnostics-14-00657-t003:** Detailed descriptions of the types of cytological samples used for cytospin preparation and immunocytochemical staining, and the corresponding diagnoses of the patients.

Diagnostic Marker	Sample Type Used for the Staining *	Patient Diagnoses
Calretinin	abdominal effusions, pleural effusions, washings from the abdomen	atypical mesothelial proliferation, adenocarcinomas
Ber-EP4	abdominal effusions, pleural effusions, washings from the pouch of Douglas	lung-, breast-, ovarian adenocarcinoma
MOC-31	abdominal effusions, pleural effusions, washings from the pouch of Douglas	lung-, breast-, ovarian adenocarcinoma
CK AE1/AE3	abdominal and pleural effusions	lung-, breast-, ovarian adenocarcinoma
CK 5/6	abdominal and pleural effusions	atypical mesothelial proliferation, lung-, breast-, ovarian adenocarcinoma
CD68	abdominal effusions, pleural effusions, washings from the pouch of Douglas	atypical mesothelial proliferation
LCA	abdominal effusions, pleural effusions, effusions in the pouch of Douglas, FNABs	inflammation, ovarian adenocarcinoma, non-Hodgkin’s lymphoma
Desmin	pleural effusions	atypical mesothelial proliferation
HBME-1	abdominal effusions, pleural effusions, abdomen washings, effusions in the pouch of Douglas	atypical mesothelial proliferation
WT1	abdominal effusions, pleural effusions, effusions in the pouch of Douglas, FNABs	atypical mesothelial proliferation, ovarian adenocarcinoma

* Five (5) cytospins per marker were used for the test protocol, and twenty cytospins were used to compare the optimal protocols (of which ten cytospins were to provide positive immunocytochemical staining results and ten, serving as negative controls, were to undergo no staining).

**Table 4 diagnostics-14-00657-t004:** Comparison of the NEQAS scores, staining results, staining intensity, background, counterstain, and morphology, evaluated on 10 positive cases stained with the optimal OptiView and EnVision protocols and the existing iView protocol for calretinin, Ber-EP4, MOC-31, CK AE1/AE3, CK 5/6, CD68, LCA, desmin, HBME-1, and WT1.

		iV	OV	EnV			iV	OV	EnV
		Median (Range)					Median (Range)		
**Calretinin**	NEQAS	14.5 (11–15)	20.0 (16–20)	16.0 (15–19)	**CD68**	NEQAS	17.0 (11–19)	18.5 (14–20)	17.0 (16–20)
	Staining	22 (4–50)	24 (5–50)	23 (5–50)		Staining	64 (28–89)	70.5 (33–89)	65.5 (14–89)
	Intensity	2 (1–2)	3 (2–3)	2.25 (2–3)		Intensity	2 (1.5–3)	3 (2–3)	2.25 (2–3)
	Background	1 (0.5–1)	0 (0–1)	1 (0–1)		Background	0 (0–1)	0 (0–1)	0 (0–0.5)
	Counterstain	0 (0–0)	0 (0–0)	0 (0–0)		Counterstain	0.1 (0–1)	0 (0–0)	0.1 (0–1)
	Morphology	0 (0–0)	0 (0–0)	0 (0–0)		Morphology	0 (0–0)	0.1 (0–1)	0 (0–0)
**Ber–EP4**	NEQAS	15.5 (13–19)	19.0 (15–20)	16.5 (15–19)	**LCA**	NEQAS	15.0 (13–17)	17.0 (13–20)	16.5 (13–19)
	Staining	39 (10–99)	36.5 (10–99)	34 (10–99)		Staining	79 (15–99)	85.5 (23–100)	89.5 (23–100)
	Intensity	2.25 (2–3)	3 (3–3)	2 (2–2.5)		Intensity	2 (2–2.5)	3 (2.5–3)	3 (2–3)
	Background	0.5 (0–1)	0 (0–1)	0 (0–1)		Background	0.25 (0–1)	0.25 (0–1.5)	1 (0–1)
	Counterstain	0.2 (0–1)	0 (0–0)	0 (0–0)		Counterstain	0 (0–0)	0 (0–0)	0.1 (0–1)
	Morphology	0.1 (0–1)	0 (0–0)	0 (0–0)		Morphology	0.3 (0–1)	0.3 (0–1)	0 (0–0)
**MOC–31**	NEQAS	16.0 (12–18)	18.5 (15–20)	17.0 (15–18)	**Desmin**	NEQAS	15.5 (12–19)	16.0 (15–20)	13.0 (4–16)
	Staining	45 (10–100)	34 (10–99)	35.5 (10–99)		Staining	2 (1–30)	7.5 (1–35)	1.5 (1–93)
	Intensity	2 (2–3)	3 (3–3)	2 (2–3)		Intensity	2 (1–3)	3 (2.5–3)	2 (0–3)
	Background	0.5 (0–1)	0 (0–1)	0.25 (0–1)		Background	0 (0–1)	1 (0–1)	1 (1–1.5)
	Counterstain	0.2 (0–1)	0 (0–0)	0 (0–0)		Counterstain	0.2 (0–1)	0.2 (0–1)	0 (0–0)
	Morphology	0.1 (0–1)	0 (0–0)	0 (0–0)		Morphology	0.1 (0–1)	0 (0–0)	0 (0–0)
**CK AE1/AE3**	NEQAS	14.5 (10–18)	18.0 (16–20)	16.0 (13–18)	**HBME–1**	NEQAS	12.0 (11–14)	13.5 (11–16)	13.8 (13–16)
	Staining	33 (3–99)	33 (3–99)	36.5 (3–99)		Staining	20.5 (3–43)	21.5 (4–43)	13 (2–33)
	Intensity	2 (2–2.5)	3 (2.5–3)	2.5 (2–3)		Intensity	2 (2–2)	3 (2–3)	2 (1–2)
	Background	1 (0–1)	0.25 (0–1)	0.5 (0–1)		Background	1.75 (1–2)	2 (1–2)	1 (0–2)
	Counterstain	0.3 (0–1)	0 (0–0)	0.5 (0–1)		Counterstain	0 (0–0)	0.2 (0–1)	0 (0–0)
	Morphology	0.2 (0–1)	0 (0–0)	0 (0–0)		Morphology	0 (0–0)	0.2 (0–1)	0 (0–0)
**CK 5/6**	NEQAS	14.5 (9–20)	18.5 (17–20)	14.5 (11–18)	**WT1**	NEQAS	15.5 (11–17)	18.5 (13–20)	13.0 (11–16)
	Staining	4.5 (1–93)	5.5 (2–43)	5.5. (2–40)		Staining	41 (3–82)	46.5 (6–82)	49 (6–82)
	Intensity	2 (1–2)	3 (3–3)	3 (2–3)		Intensity	2 (1.5–2)	3 (3–3)	2 (2–2.5)
	Background	1 (0–1)	0 (0–1)	1 (0–2)		Background	0.5 (0–1)	0.25 (0–1)	1 (1–2)
	Counterstain	0.1 (0–1)	0 (0–0)	0.2 (0–1)		Counterstain	0.3 (0–1)	0.2 (0–1)	0.3 (0–1)
	Morphology	0.1 (0–1)	0.1 (0–1)	0.2 (0–1)		Morphology	0.1 (0–1)	0.3 (0–1)	0.1 (0–1)

Abbreviations: iV—iView; OV—OptiView; EnV—EnVision FLEX.

**Table 5 diagnostics-14-00657-t005:** *p*-values for the comparison of all evaluated parameters described in Table 4.

		iV/OV	iV/EnV	Ov/EnV			iV/OV	iV/EnV	Ov/EnV
		*p* Value					*p* Value		
**Calretinin**	NEQAS	<0.001	0.002	0.002	**CD68**	NEQAS	NS	NS	NS
	Staining	NS	NS	NS		Staining	NS	NS	NS
	Intensity	0.003	0.031	0.019		Intensity	0.012	NS	NS
	Background	0.010	NS	NS		Background	NS	NS	NS
	Counterstain	NS	NS	NS		Counterstain	NS	NS	NS
	Morphology	NS	NS	NS		Morphology	NS	NS	NS
**Ber–EP4**	NEQAS	0.001	NS	0.004	**LCA**	NEQAS	NS	NS	NS
	Staining	NS	NS	NS		Staining	0.014	0.011	0.017
	Intensity	0.012	NS	0.004		Intensity	0.005	0.007	NS
	Background	NS	NS	NS		Background	NS	NS	NS
	Counterstain	NS	NS	NS		Counterstain	NS	NS	NS
	Morphology	NS	NS	NS		Morphology	NS	NS	NS
**MOC–31**	NEQAS	0.017	NS	NS	**Desmin**	NEQAS	NS	0.036	0.005
	Staining	NS	NS	NS		Staining	0.029	NS	0.029
	Intensity	0.007	NS	0.020		Intensity	0.013	NS	0.009
	Background	NS	NS	NS		Background	NS	0.007	NS
	Counterstain	NS	NS	NS		Counterstain	NS	NS	NS
	Morphology	NS	NS	NS		Morphology	NS	NS	NS
**CK AE1/AE3**	NEQAS	0.004	0.005	0.004	**HBME–1**	NEQAS	NS	0.019	NS
	Staining	NS	NS	NS		Staining	NS	NS	0.028
	Intensity	0.004	0.012	0.045		Intensity	0.006	NS	0.006
	Background	NS	NS	NS		Background	NS	NS	NS
	Counterstain	NS	NS	0.044		Counterstain	NS	NS	NS
	Morphology	NS	NS	NS		Morphology	NS	NS	NS
**CK 5/6**	NEQAS	0.002	NS	0.003	**WT1**	NEQAS	0.007	0.030	0.002
	Staining	NS	NS	NS		Staining	NS	NS	NS
	Intensity	0.006	0.012	NS		Intensity	0.003	NS	0.004
	Background	NS	NS	NS		Background	NS	0.012	0.013
	Counterstain	NS	NS	NS		Counterstain	NS	NS	NS
	Morphology	NS	NS	NS		Morphology	NS	NS	NS

Abbreviations: iV—iView; NS, non–significant; OV—OptiView; EnV—EnVision FLEX.

## Data Availability

Data will be available after considering the aim of further use.

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
