# Peer review of "Validation and Implementation of OptiView and EnVision FLEX Detection Systems for Immunocytochemical Staining Protocols of the Ten Most Commonly Used Diagnostic Markers in Routine Cytopathological Practice"

_diagnostics, 2024, doi:10.3390/diagnostics14060657_

Round 1

Reviewer 1 Report

Comments and Suggestions for Authors

General remarks

The study is well-designed and also well written. Here, the authors selected certain antibodies and established the immunocytochemistry ( ICC) protocol on two important techniques: Opti view (OV) and Envision FLEX (ENV). The results of the ICC protocol were comparable.

This study is well-written, with good figures and illustrations.

The discussion and conclusions are also well-written.

Author Response

We thank Reviewer 1 for taking the time to review our manuscript and for the positive report and supportive comments.

Reviewer 2 Report

Comments and Suggestions for Authors

The article of Dremelj et al “Validation and implementation of OptiView and EnVision FLEX detection systems for immunocytochemical staining protocols of the ten most commonly used diagnostic markers in cytopathological routine practice” is devoted to optimization, validation, and implementation of immunocytochemistry protocols using OptiView (OV) and EnVision FLEX (EnV) detection systems, as well as comparison the results with those obtained using the iView detection system (iV) forced many cytopathology laboratories.

This is a well-written and well-illustrated manuscript, but I have some comments and suggestions regarding this article.

Major points:

1. The main limitation of this study is that the staining protocols were developed for only 10 markers and only for cells obtained from effusion puncture/abdominal washings. These restrictive conditions (the number of markers and the origin of cells) significantly reduce scientific and practical value of the study. Could you please include a “Limitations of the study” section in the manuscript?

2. It would be useful to slightly expand Section 2.1 by including information concerning the diagnosis of samples (non-neoplastic conditions or malignant neoplasms) used in the study.

3. Why were samples of abdominal effusions/washings chosen for validation and implementation of staining protocols? Could you comment on the choice of research object?

4. Could it turn out that staining for the same markers of cells of a different origin will require different conditions for optimal staining results? Please, comment this point.

Minor point:

The abstract should be a single paragraph and should follow the style of structured abstracts, but without headings. Please, correct.

Author Response

We thank Reviewer 2 for taking the time to review our manuscript. Our responses can be found below and the relevant revisions/corrections are tracked in the re-submitted manuscript file.

Open Review

(x) I would not like to sign my review report

( ) I would like to sign my review report

Quality of English Language

(x) I am not qualified to assess the quality of English in this paper

( ) English very difficult to understand/incomprehensible

( ) Extensive editing of English language required

( ) Moderate editing of English language required

( ) Minor editing of English language required

( ) English language fine. No issues detected

Yes      Can be improved       Must be improved      Not applicable

Does the introduction provide sufficient background and include all relevant references?

( )         (x)        ( )        ( )

Our comment: the main aim of our study was to compare three different detection systems (developed primarily for histology) for immunocytochemical staining (which is not standardized among cytopathology laboratories) and we believe we have addressed this subject in the introduction.

Are all the cited references relevant to the research?

(x)        ( )         ( )        ( )

Is the research design appropriate?

(x)        ( )         ( )        ( )

Are the methods adequately described?

(x)        ( )         ( )        ( )

Are the results clearly presented?

( )         (x)        ( )        ( )

Our comment: since the main aim of our study was to compare three different detection systems in cytology samples for the 10 most commonly used diagnostic markers in cytology laboratories, we believe that showing the differences in staining results of the different parameters for each marker as well as the parallel comparison of the staining results of the optimally selected staining protocols with all three detection systems for each marker fulfills the purpose of our study.

Are the conclusions supported by the results?

( )         ( )         (x)        ( )

Our comment: we believe that we have improved the clarity of our conclusions by providing additional explanations regarding the inclusion criteria for the patient samples. We have also addressed the limitations of our study and provided a detailed explanation based on our many years of cytologic experience, including unpublished data related to the application of staining protocols to samples from origin other than those originally used in the study.

Comments and Suggestions for Authors

The article of Dremelj et al “Validation and implementation of OptiView and EnVision FLEX detection systems for immunocytochemical staining protocols of the ten most commonly used diagnostic markers in cytopathological routine practice” is devoted to optimization, validation, and implementation of immunocytochemistry protocols using OptiView (OV) and EnVision FLEX (EnV) detection systems, as well as comparison the results with those obtained using the iView detection system (iV) forced many cytopathology laboratories.

This is a well-written and well-illustrated manuscript, but I have some comments and suggestions regarding this article.

Major points:

  1. The main limitation of this study is that the staining protocols were developed for only 10 markers and only for cells obtained from effusion puncture/abdominal washings. These restrictive conditions (the number of markers and the origin of cells) significantly reduce the scientific and practical value of the study. Could you please include a “Limitations of the study” section in the manuscript?

Thank you for your input. We have integrated this section into the conclusion chapter.

  1. It would be useful to slightly expand Section 2.1 by including information concerning the diagnosis of samples (non-neoplastic conditions or malignant neoplasms) used in the study.

Thank you for the suggestion; we have extended Section 2.1 with additional information on patient inclusion criteria. However, it is important to mention that is complex to explain the variety of diagnoses among the patients included in the study, since it was not our primary focus. We have included patients who, based on their cytopathological diagnosis and cytopathologist’s experience to predict for which of the selected markers in the study the sample is expected to be positive or negative, regardless of whether it is non-neoplastic or malignant. For example, non-neoplastic effusions were expected to be positive for calretinin and HMBE-1 (mesothelial markers) and negative for Ber-EP4 and MOC-31 (tumor markers), which means those samples could be used as positive cases for calretinin and HMBE-1 and negative controls for Ber-EP4 and MOC-31. Furthermore, some malignant neoplasms, ex. those originating from the lung, were expected to be positive for instance for CK5/6 but negative for WT1 (a marker of ovarian origin and mesothelial origin) and could be used as positive cases for CK5/6 and negative controls for the WT1. Therefore, the expected positivity/negativity of the samples for the markers was more important for the study than the type of diagnosis of the patients.

  1. Why were samples of abdominal effusions/washings chosen for validation and implementation of staining protocols? Could you comment on the choice of research object?

As mentioned in the study design section of the manuscript, we only used leftover cytological samples of patients whose samples were initially collected for routine diagnostic purposes at our Institute of Oncology Ljubljana. The reason why we chose to perform the testing on abdominal effusions/washings instead of fine-needle aspiration biopsy (FNAB) samples from lymph nodes was the greater availability of leftover cell material from effusions/washings, allowing us to prepare 20-30 cytospins beside the cytospins that were initially used for diagnostic examination; from FNAB samples in most of the cases we have limited amount of sample material, enough for a total of 10-12 cytospins, most of which are usually used for diagnostic examination. Our study was committed to good ethical practice to avoid subjecting patients to additional FNAB sampling since we have sufficient leftover material to provide from the effusions/washings. Once again, our primary purpose was to investigate the staining parameters using OptiView and Envision FLEX detection systems for samples that are positive for the selected diagnostic markers in comparison with the iView, which showed that target retrieval and amplification are rather needed to create new staining protocols, and it is our main information useful for implementing diagnostic markers with these two detection systems.

  1. Could it turn out that staining for the same markers of cells of a different origin will require different conditions for optimal staining results? Please, comment this point.

Our Cytology Department has experience of more than 70 years, performing immunocytochemistry since 1986 and using automatic immunostainers since 2002 which have resulted with more than 8,000 stainings per year in the last half decade (source: https://www.youtube.com/watch?v=NdGEoFedMCk). As mentioned above, our long-term practice has shown that in general, FNAB samples do not provide enough leftover material for testing and we rather perform testing on effusion samples. Our unpublished, in-house follow-up analysis has confirmed that each time we implement optimized protocols into our everyday routine practice for samples of different origins, we do not observe variations/deviations in the staining reaction of all types of diagnostic samples. We control this action by comparing the staining reaction of each sample (regardless of the origin) with a positive and negative control (cytospins), which are part of each staining round. On the other side, for samples, such as FNAB samples for lymphoma diagnosis we prefer using flow cytometric analysis, and for very rare soft tissue samples, such as sarcomas, and salivary gland samples we do prospective marker optimizations since otherwise we could not provide suitable samples for testing and optimization.

Minor point:

The abstract should be a single paragraph and should follow the style of structured abstracts, but without headings. Please, correct.

Thank you for your observation. We have made the necessary corrections and believe we accurately performed the intended request.

Round 2

Reviewer 2 Report

Comments and Suggestions for Authors

The authors took into account all suggestions. The manuscript may be published in its present form.